# From crisis to recovery: Exploring the demand surge for mental health services in Alberta, Canada—A document-based policy analysis with an illustrative supply–demand simulation (2023–2024)

Kola Adegoke[1,2]*, Abimbola Adegoke[1,2], Deborah Dawodu[3], Ayoola Bayowa[4], Akorede Adekoya[5], Temitope Kayode[1], Mallika Singh[1], Olajide Alfred Durojaye[6], Abiodun Isola Aluko[7], Adeyinka Adegoke[8]

1 School of Health Sciences and Practice, New York Medical College, Valhalla, New York, United States of America, 2 Department of Health & Biomedical Sciences, College of Health Professions, University of Texas, Rio Grande Valley, One West University Blvd, Brownsville, Texas, United States of America, 3 McWilliams School of Biomedical Informatics, UTHealth Houston, Houston, Texas, United States of America, 4 Bob Gaglardi School of Business and Economics, Thompson Rivers University, Kamloops, British Columbia, Canada, 5 Department of Virtual Health, Fraser Health, Surrey, British Columbia, Canada, 6 Department of Family Medicine, Carleton Medical Clinic, St. Albert, Alberta, Canada, 7 Program Management Office (PMO), CASA Mental Health (for Children & Families in Alberta), Edmonton, Alberta, Canada, 8 Department of Business Administration, Northern Alberta Institute of Technology, Edmonton, Alberta, Canada

* kadegoke@student.touro.edu

## Abstract

COVID-19 coincided with increased mental health needs in Alberta, Canada, intensifying pre-existing access gaps and service strain. Alberta responded with publicly funded interventions spanning digital care, youth-focused services, and recovery-oriented programs. To evaluate Alberta's system-level response to pandemic-related increases in mental health help-seeking/service uptake using a health economics and policy lens. We extracted empirically reported program delivery outputs from the 2023–2024 Alberta Mental Health and Addiction Annual Report. We used a simulation calibrated to reported trends to examine directional changes in help-seeking (demand), service capacity (supply), and the modeled equilibrium quantity under a zero-copayment design. Empirically reported outputs indicate that delivery met or exceeded planned/funded milestones for CASA Mental Health, VODP, and tele-mental health, while recovery communities reflected phased implementation. In the illustrative simulation, the demand-implied volume increases from 60 to 87 services/month, but delivered volume is capacity-constrained at 78 services/month (implying ~9 services/month unmet demand), while a unit-cost proxy is held constant for visualization (not an observed market price or patient copayment). Alberta's response illustrates how coordinated, publicly funded capacity expansion and access-oriented policies can support service delivery during system shocks; the

**Data availability statement:** All materials supporting the findings of this study, including the extracted summary inputs from public reports, README, CHEERS 2022 guideline's checklist, modeling assumptions/parameter table, analysis workbook(s), and figure files, are available on OSF: Registration DOI 10.17605/OSF.IO/6DBFU, Associated project https://osf.io/e6n7w Underlying sources remain publicly accessible through the Government of Alberta and Alberta Health Services repositories, as well as the cited literature.

**Funding:** The authors received no specific funding for this work.

**Competing interests:** The authors have declared that no competing interests exist.

model also highlights that if capacity growth lags demand growth, unmet need may persist even under zero copayment.

## Summary

During the COVID-19 pandemic, more people in Alberta needed mental health and addiction support. Alberta responded by expanding publicly funded services, including youth programs (CASA Mental Health classrooms), same-day virtual opioid treatment (VODP), tele-mental health, and recovery communities. Using Alberta's 2023–2024 public reporting, we summarize what programs were planned to deliver and what they actually delivered. We also use a simple, illustrative supply–demand simulation (not direct price measurement) to show how increasing service capacity at the same time as help-seeking rises can increase the number of services delivered while keeping user costs at zero. Overall, Alberta's approach shows how coordinated public investment and service redesign can support access during a system shock. To sustain progress, Alberta will likely need continued workforce investment, strong digital access in rural areas, and equity-focused service design for youth and Indigenous communities.

## 1. Introduction

The COVID-19 pandemic coincided with increased mental health needs, including anxiety, depression, and substance use across age groups, with children and youth disproportionately affected [1]. In Canada's publicly funded health system, access to mental health services remains uneven across provinces due to structural inequities, governance differences, and workforce constraints [2].

Despite extensive documentation of COVID-era mental health needs and policy responses, relatively few studies quantify how publicly funded mental health systems adjust to concurrent demand shocks and supply expansions using an explicit economic framing. Using Alberta's 2023–2024 administrative reporting, we apply a system-level supply–demand model to translate policy actions into interpretable implications for access and utilization.

Alberta is a useful case because of its decentralized delivery structure and rapid implementation of publicly funded interventions between 2020 and 2024. Key initiatives included the Virtual Opioid Dependency Program (VODP), expanded CASA Mental Health classrooms for high-needs youth, and the creation of recovery communities with additional publicly funded residential capacity [3]. These investments were paired with efforts to expand digital access and improve mental health literacy, which may have influenced both service demand (help-seeking) and system capacity (service availability).

This study examines Alberta's system-level response through a health economics and policy lens using two complementary components: (1) **empirically reported program delivery outputs** from the 2023–2024 Alberta Mental Health and Addiction Annual Report and (2) **an illustrative supply–demand simulation** calibrated to

reported trends to assess directional shifts in demand, capacity, and the modeled equilibrium quantity under a zero-copayment design. The aim is to clarify how coordinated public interventions may support service delivery during system shocks and to identify policy-relevant considerations for equity and sustainability, without estimating cost-effectiveness outcomes.

## 3. Literature review and theoretical framework

### 3.1. Supply and demand in mental healthcare markets

Mental healthcare operates in a constrained submarket with persistent market failures: supply is limited by workforce shortages and geographic maldistribution, while demand is shaped by stigma, diagnostic uncertainty, and gatekeeping. In publicly financed systems, standard "market equilibrium" concepts require adaptation because third-party payment, non-transparent pricing, and information problems weaken the link between patient prices and utilization [4]. In this paper, we use "equilibrium" in a limited sense to describe the relationship between system capacity (service availability) and help-seeking/service uptake under a zero-copayment design.

During the COVID-19 period, Alberta experienced increased help-seeking and service uptake—particularly among youth and rural populations—alongside capacity expansion through digital modalities and targeted investments [2,3]. International calls following COVID-19 similarly emphasize strengthening digital access, expanding community-based care, and rebalancing hospital-centric models to improve continuity and equity [5]. Alberta's intervention mix provides a valuable case for examining how policy instruments may influence both capacity and uptake within a decentralized provincial system.

### 3.2. Economic frictions relevant to mental health policy

**Moral hazard.** When out-of-pocket costs are low, utilization can increase, including for lower-acuity needs, potentially straining capacity if not managed [4]. In Alberta, publicly funded services (including VODP and youth-focused programs such as CASA) minimize user costs. They may increase use, making triage, referral pathways, and eligibility criteria important for balancing access and capacity [3]. Evidence from the RAND Health Insurance Experiment demonstrates that cost-sharing affects utilization across service types, including mental health, underscoring the behavioral implications of insurance design [6].

**Adverse selection.** Mental health systems can face selection pressures when higher-need individuals disproportionately seek intensive services, potentially increasing resource concentration and wait times [4]. In Alberta, targeted programs for high-need youth and other priority groups may concentrate demand within specific service lines, reinforcing the need to match the workforce and infrastructure to population needs [1,3].

**Provider-induced demand (PID).** Clinician influence on service intensity can contribute to overuse, especially where payment or performance incentives reward volume. Although PID is typically less prominent in publicly funded systems, rapid scale-up of virtual and community-based services can alter incentives and monitoring needs. Where standardized protocols and performance monitoring are in place, they may mitigate unnecessary repeat visits while protecting access [4,7,8].

**Information asymmetry.** In mental health, low health literacy, stigma, and limited outcome transparency can distort help-seeking and reduce accountability. Alberta has pursued outreach and culturally adapted services, yet gaps in real-time outcome and equity measurement limit system learning and targeted improvement [2,3]. Without accessible performance information, both underuse (among underserved groups) and congestion (in high-visibility services) can persist.

### 3.3. Alberta's policy–economic positioning

From a health economics perspective, Alberta's COVID-era response can be interpreted as an attempt to address multiple market failures simultaneously, improving access under a zero-copayment design while expanding capacity through digital delivery and recovery-oriented infrastructure [3,4]. In this study, we avoid interpreting modeled values as observed

prices. Instead, we use an illustrative simulation to examine how coordinated shifts in capacity and uptake could increase equilibrium quantity under a publicly financed system, while highlighting equity-relevant constraints where Alberta-specific elasticities and marginal costs are unavailable.

### 3.4. Comparative policy context

To situate Alberta's approach, we provide a concise comparison with selected jurisdictions that also expanded tele-mental health and community-based services during COVID-19. These comparisons are used to frame plausibility and policy relevance rather than to generate comparative effectiveness estimates.

**Ontario (Canada).** Ontario's more centralized planning and performance infrastructure, through Ontario Health's Mental Health and Addictions Centre of Excellence, which establishes provincial oversight, common performance indicators, and shared infrastructure, provides a contrast to Alberta's decentralized, programmatic expansion [8].

**Australia and the United Kingdom.** Both contexts expanded telehealth and remote access during COVID-19, but implementation barriers (digital literacy, uneven access to video platforms, and persistent uptake inequities) were reported, especially among underserved populations [9,10].

**United States and European Union.** Insurance design and reimbursement structures shaped utilization and equity, with cost-sharing and fragmented coverage creating distinct barriers and incentives compared with single-payer settings. However, persistent treatment gaps have been documented in multiple regions, highlighting that coverage expansion alone may not close disparities without culturally relevant services and targeted outreach [11–13].

Overall, Alberta's experience is consistent with a broader post-pandemic policy trend toward digital access and community-based recovery supports. The remaining challenge is aligning capacity expansion with equity goals and transparent performance measurement in a decentralized system.

Taken together, this literature and policy context motivates a two-part analytic approach. First, we summarize empirically reported program delivery outputs from Alberta's 2023–2024 Mental Health and Addiction Annual Report, and aggregate public reporting to describe implementation performance during the reporting period. Second, because Alberta-specific elasticities and unit costs are not consistently available in public sources, we use an illustrative supply–demand simulation calibrated to reported trends to examine the directional implications of simultaneous shifts in service capacity and help-seeking under a zero-copayment design. The Methods section below specifies data sources, parameter assumptions, and the uncertainty analyses used to ensure that modeled outputs are interpreted transparently as simulations rather than observed market prices or utilization counts.

## 4. Methodology

### 4.1. Design

The study used a document-based policy evaluation, combined with an illustrative health economics simulation, to examine Alberta's system-level response to pandemic-era increases in mental health service need. We applied a supply–demand framework to interpret how publicly funded interventions—particularly CASA Mental Health classroom expansion, the Virtual Opioid Dependency Program (VODP), and recovery communities- could shift service capacity and utilization during the 2023–2024 fiscal year [3]. The analysis is intended to clarify directional system dynamics (capacity expansion and access) rather than estimate patient-level clinical effects. The economic component is an illustrative simulation designed to transparently communicate mechanisms and key drivers, consistent with good modeling practice guidance [14]. Because the objective was explanatory (mechanism-focused), no formal economic evaluation (e.g., cost-effectiveness/cost-utility), resource valuation, or health outcome valuation (e.g., QALYs) was undertaken; instead, the model is used to explore plausible directional implications under limited parameter availability, consistent with the role of early/illustrative modeling in informing decisions under uncertainty in innovations [15].

### 4.2. Data sources

Empirical inputs were extracted from the Government of Alberta's *2023–2024 Mental Health and Addiction Annual Report* [3], including program delivery volumes, planned versus actual service outputs, indicators of service deployment, and infrastructure and system investment descriptions. We used Russell et al. (2024) to contextualize utilization trends in youth mental health care during the pandemic period in Alberta [2]. Additional sources were used for conceptual framing and parameterization when Alberta-specific estimates were unavailable, including foundational health economics references [4], evidence on utilization responses to cost-sharing from the RAND Health Insurance Experiment [6], and the supplier-induced demand literature, which is used to motivate sensitivity analyses around provider behavior [7].

### 4.3. Assumptions and parameters

The simulation required simplified assumptions to translate public reporting into an interpretable supply–demand representation. Key assumptions were:

1. **Capacity expansion modeled proportionally:** The reported scale-up in service capacity during 2023–2024 was modeled as a proportional rightward shift in supply, applied uniformly across major population groups (e.g., rural, youth, and Indigenous), given the absence of stratified administrative data in public sources [3].

2. **Zero user copayment by design:** Because Alberta's publicly funded services maintain **zero user copayment**, the user price was treated as 0 CAD. For illustrative purposes only, we used a **unit cost proxy (CAD/service)** that is not an observed market price and is not a patient copayment [6].

3. **Quality held constant across regions and modalities:** In the absence of comparable public outcome measures by geography or modality, we assumed no systematic quality differences across regions and delivery channels, consistent with published service protocols and clinical guidance [16].

4. **Alberta Specific Estimates:** Where Alberta-specific estimates of marginal cost and price elasticity were unavailable, we used Canadian benchmarking sources to contextualize spending levels and plausible resource ranges (e.g., CIHI NHEX) [17]. Plausible elasticity ranges were informed by evidence on utilization responses to cost-sharing [6] and aligned with standard health economics guidance [4].

The demand shift was calibrated to $\Delta D = +27$ services/month (bounds 20–35), based on Russell et al. (2024), which reported a +27.24/month increase in youth GP mental health visits per 100,000 after the initial pandemic drop; this was used as an illustrative demand-shift magnitude [2].

A complete parameter table (base values, bounds, units, and source attribution) is provided in S1 File and the OSF repository to support transparency and reproducibility.

### 4.4. Supply–demand simulation modeling

We used an illustrative supply–demand simulation to represent system-level changes in Alberta's publicly funded mental health services during the 2023–2024 fiscal year. The analytic perspective was the provincial public payer/system perspective, focusing on publicly funded capacity and program outputs rather than patient-level outcomes. The time horizon was 12 months; therefore, discounting was not applied. Monetary values are reported in **Canadian dollars (CAD)** and reflect the **2023–2024 reporting period**; no inflation or purchasing power parity adjustments were applied.

**4.4.1. Supply curve shift (modeled).** Supply expansion was modeled as a rightward shift in the supply curve, reflecting reported increases in service capacity attributable to workforce expansion, digital delivery modalities, and recovery-oriented infrastructure investments documented in Alberta's 2023–2024 Annual Report [3], supported by standard health economics framing [4].

**4.4.2. Demand curve shift (modeled).** Demand growth was modeled as a rightward shift in the demand curve, reflecting increased help-seeking associated with pandemic-related distress, destigmatization, and improved accessibility through interventions such as same-day virtual opioid dependency care (VODP) and youth-focused programming [2,3].

**4.4.3. Interpretation of "price" and equilibrium effects.** Because Alberta's publicly funded mental health services operate under a **zero-user copayment** policy, the **user price is treated as 0 CAD**. To express equilibrium relationships in interpretable units, we used a **unit-cost proxy (CAD/service)** for modeling convenience. This unit cost proxy is **not an observed market price** and should not be interpreted as a measured transaction price; it is used only to illustrate equilibrium mechanics in a publicly financed setting.

Model outputs in Tables and Figures are **synthetic simulation outputs calibrated to reported trends** and explicitly labeled as modeled results to avoid confusion with empirically observed data. Empirical program delivery metrics, extracted directly from public reporting (e.g., planned versus actual services delivered), are reported separately in the Results section as observed outputs.

**4.4.4. Software and reproducibility.** All calculations and figure/tabular outputs were produced in **Microsoft Excel for Mac (Version 16.104 [build 25121423])**. Data sources, parameter values (with bounds), assumptions, and sensitivity/scenario checks are archived in the OSF repository: **Registration DOI:10.17605/OSF.IO/6DBFU**, associated project: https://osf.io/e6n7w, along with reproduction instructions (S1 Text, S1 File).

**4.4.5. Reporting and transparency.** We used the CHEERS 2022 statement to guide transparent reporting of methods and model-related assumptions where applicable; however, this study is a document-based policy analysis with an illustrative simulation and does not report a full health economic evaluation (e.g., costs and consequences comparison, ICERs/QALYs). Therefore, CHEERS items relevant to context, methods, assumptions, uncertainty, and reporting were addressed, and non-applicable items were noted in the completed checklist (S1 Checklist, OSF) [19].

**4.4.6. Ethics approval and consent to participate.** This study used publicly available, aggregate, non-identifiable data from the Government of Alberta and Alberta Health Services, as well as published literature. No individual-level human participant data were collected or accessed, and there was no interaction with human participants. Therefore, institutional ethics review and informed consent were not required under applicable regulations and institutional policies.

## 5. Results

This section distinguishes **empirical results** (directly observed program outputs from public reports) from **modeled results** (synthetic supply–demand simulations calibrated to reported trends). Modeled values should not be interpreted as directly observed market prices or utilization counts unless explicitly identified as empirical.

### 5.1. Empirical results: observed program delivery indicators (2023–2024)

Actual delivery included 96 students in CASA classrooms (exceeding the 6 planned additions), 7,217 new VODP admissions (with expansions funded but no specific numeric target stated), 219 clients in recovery communities (toward 700 beds planned), and ~12,300 tele-mental health sessions (with 14 counsellors added).

As shown in **Table 1** and **Fig 1**, actual delivery exceeded planned targets for CASA, VODP, and tele-mental health, while recovery communities delivered slightly below plan.

These empirical delivery outputs indicate implementation performance during the reporting period and provide the observed context for the modeled equilibrium illustrations presented below.

### 5.2. Modeled results: supply–demand simulations (illustrative equilibrium shifts)

We next present modeled supply–demand simulations calibrated to reported trends to illustrate how simultaneous shifts in service capacity and demand could yield a higher equilibrium quantity under a **zero-copayment policy design**. These

**Table 1. Planned/funded expansions and reported reach/activity by program in Alberta (2023–2024) (empirical).**

| Program Type | Planned/Funded Expansions (as reported) | Actual Services Delivered (as reported) | Source Notes (definition/pages) |
|---|---|---|---|
| CASA Mental Health (Youth) | Establish 6 additional classrooms; $30M funding for expansion toward 60 classrooms by 2026; $14M for CASA Houses to serve up to 324 youth/year once operational | 6 classrooms added (total 8 operational); 96 students supported in classrooms; 647 unique clients in Core programs (417 new); >690 referrals | Pages 43–47: Focus on classrooms and Core programs; no session-based metric, so use clients/students. |
| VODP (Addiction Care) | $11.2M funding for continued operation and expansion to 24/7 in 2024–25 | 7,217 new admissions; 6,595 active clients; 7,938 referrals; 1,987 transitions to other care | Pages 52–54: Admissions and clients; increase from prior years noted. |
| Recovery Communities | Establish 11 sites (5 Indigenous partnerships); add ~700 beds to serve up to 2,000 Albertans/year; $24M (Red Deer), $19M (Lethbridge) | 2 sites opened (Red Deer: 75 beds, 145 clients; Lethbridge: 50 beds, 74 clients); total 219 people served; 2 more under construction | Pages 29–31, 35–36: Clients served; beds operational by March 2024. |
| Tele-Mental Health (Counselling Alberta) | $3.7M funding for expansion (toward $6.9M over 3 years); add 14 counsellors; expand to rural/in-person sites | ~12,300 sessions; >2,150 unique clients (95% reported improvement); services to 13 additional communities | Pages 53–54: Sessions and clients; significant increase from 2022–23. |

**Source:** Government of Alberta, *2023–2024 Mental Health and Addiction Annual Report*. [3]

**Note:** "Services" refers to the program delivery unit reported by the source (e.g., clients/admissions/sessions/beds, as defined in the Annual Report).

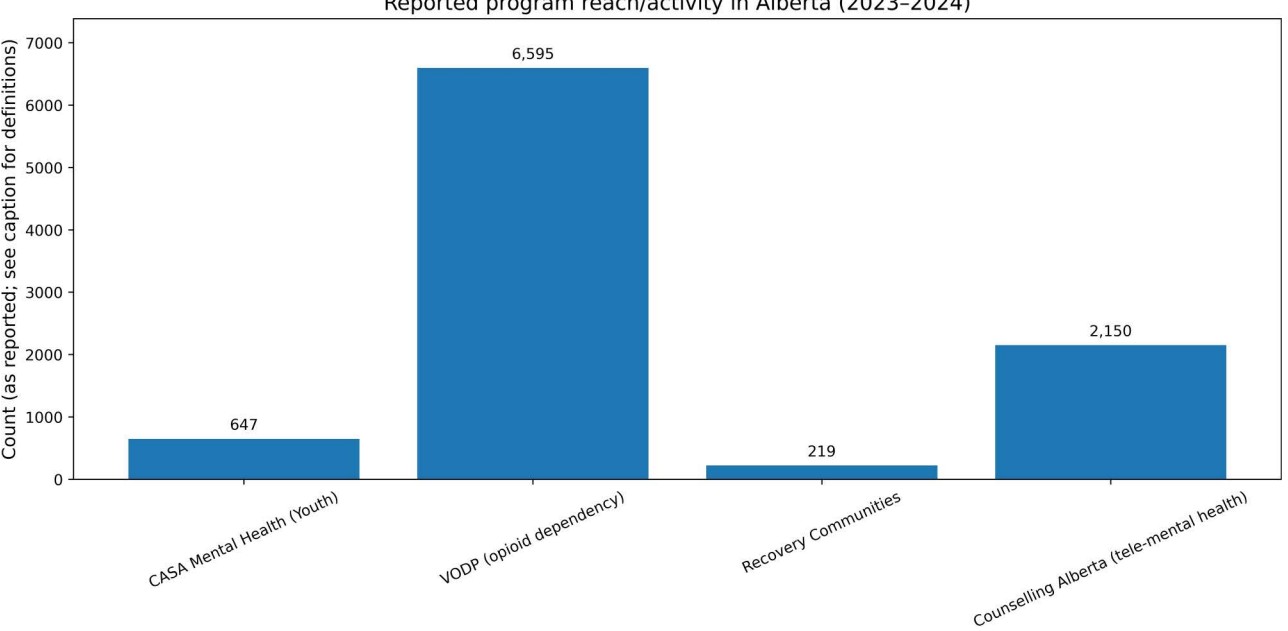

**Fig 1. Reported program reach/activity in Alberta (2023–2024).** This figure summarizes empirically reported planned targets and actual services delivered for key publicly funded programs during the 2023–2024 fiscal year. Actual delivery exceeded planned targets for CASA, VODP, and tele-mental health, while recovery communities fell slightly below plan, which may reflect implementation timing or capacity constraints. **Source:** Government of Alberta Annual Report (2023–2024) [3].

modeled outputs are illustrative simulations intended to show directional effects and equilibrium mechanics; they are not direct observations of market prices or system-wide utilization.

**5.2.1. Modeled demand shift.** Modeled demand increased in response to improved awareness, reduced stigma, and expanded access (including CASA and VODP), as reflected by a rightward shift from $D_1$ to $D_2$. As shown in Table 2 and Fig 2, the illustrative simulation depicts a rightward shift in modeled demand ($D_1 \rightarrow D_2$), consistent with increased

**Table 2. Modeled demand shift (D₁→D₂) calibrated to observed post-COVID utilization trend (illustrative).**

| Price proxy (CAD/service) | Quantity demanded (pre-COVID) | Quantity demanded (post-COVID) |
|---|---|---|
| 250 | 20 | 47 |
| 200 | 40 | 67 |
| 150 | 60 | 87 |
| 100 | 80 | 107 |
| 50 | 100 | 127 |

Values are simulated to illustrate directional demand shifts calibrated to reported trends; they are not directly observed utilization counts.

**Source:** Modeled using trends reported in Alberta's 2023–2024 Mental Health and Addiction Annual Report [3] and Alberta youth utilization evidence [2].

Note: Quantities are illustrative "services/month" units used to visualize the directional demand shift. The post-shift column adds +27 services/month at each price-proxy level, reflecting the base-case demand increase calibrated from Russell et al. (2024) utilization trends [2]. The "price proxy" is a visualization device (not an observed market price, nor a patient copayment under Alberta's zero-copayment design).

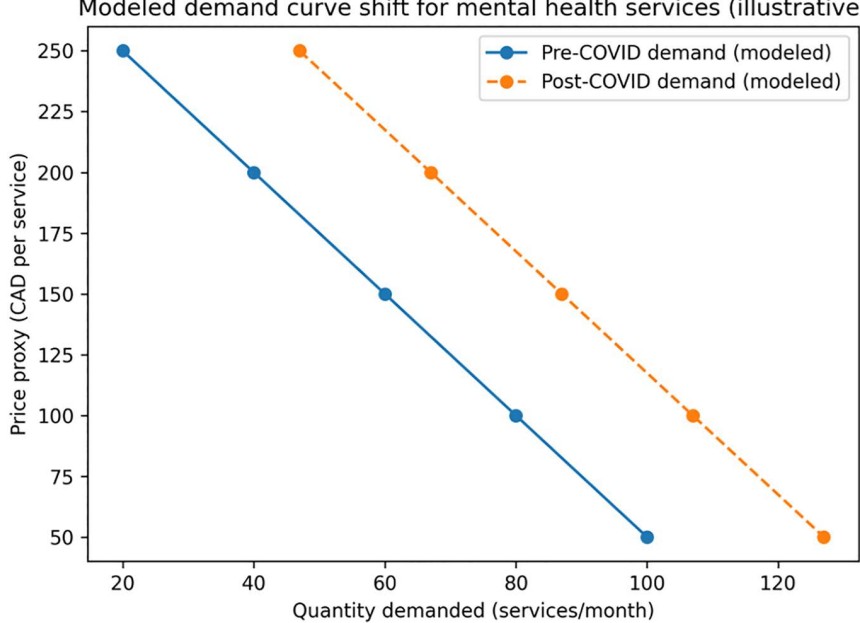

**Fig 2. Modeled demand curve shift for mental health services in Alberta (illustrative).** Note: This figure shows an **illustrative, modeled** rightward shift in demand, calibrated to reported trends in help-seeking and service uptake during the pandemic. The vertical axis reports a **unit-cost proxy (CAD/service)** used solely to visualize equilibrium mechanics in interpretable units; it is **not an observed market price** and does not represent patient copayment (user copayment is 0 CAD by policy design). **Source:** Simulation calibrated to reported trends using Alberta public reporting and supporting literature. [2–4].

help-seeking and improved access reported during the period. The model and calibration are based on reported trends rather than observed utilization counts.

**5.2.2. Modeled supply shift.** Modeled supply increased following capacity expansion through workforce measures, telehealth scale-up, and recovery-oriented infrastructure, represented as a rightward shift from S₁ to S₂. As shown in Table 3 and Fig 3, the illustrative simulation depicts a rightward shift in modeled supply (S₁→S₂), consistent with reported

**Table 3. Modeled capacity (supply) expansion (S₁ → S₂) under a proportional capacity increase (illustrative).**

| Price proxy (CAD/service) | Quantity supplied (pre-COVID) | Quantity supplied (post-COVID) |
|---|---|---|
| 50 | 20 | 26 |
| 100 | 40 | 52 |
| 150 | 60 | 78 |
| 200 | 80 | 104 |
| 250 | 100 | 130 |

Values are simulated to illustrate capacity expansion calibrated to reported infrastructure/workforce expansions; they are not directly observed system-wide service counts.

**Source:** Modeled using capacity and implementation trends reported in Alberta's 2023–2024 Mental Health and Addiction Annual Report [3] and Alberta Health Services reporting [16].

Quantities are illustrative "services/month" units used to visualize capacity expansion. The post-shift column applies a + 30% proportional capacity increase (post = pre × 1.30), consistent with the base-case system expansion assumption drawn from public reporting [3]. Values are rounded to whole services. The "price proxy" is used only to display the curve; it is not a patient price.

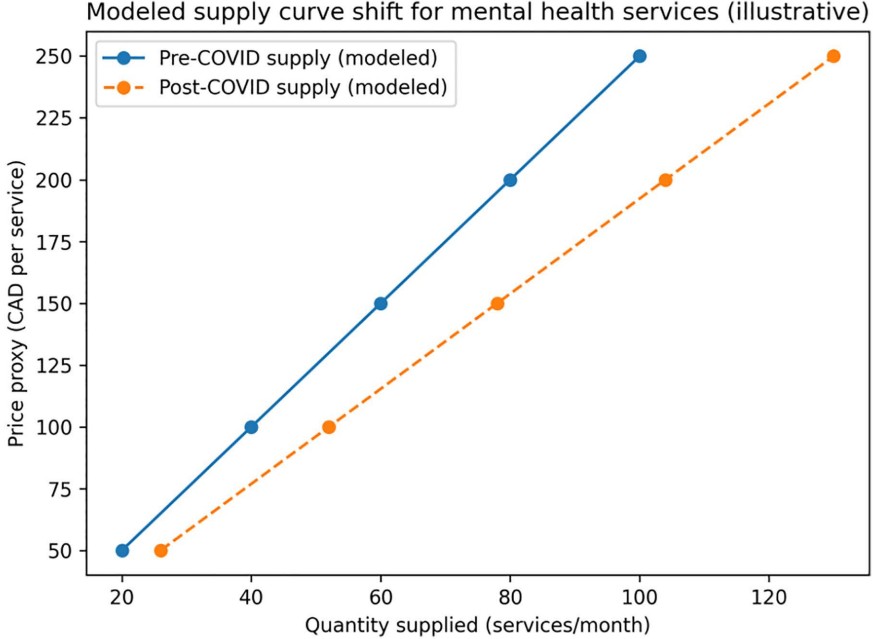

**Fig 3. Modeled supply curve shift for mental health services in Alberta (illustrative).** Note: This figure illustrates a modeled rightward shift in supply, representing expanded service capacity driven by workforce growth, telehealth scale-up, and recovery-oriented infrastructure investments during 2023–2024. The vertical axis shows a unit-cost proxy (CAD/service) used solely for visualization; it is not an observed market price and does not reflect patient copayments (user copayments are 0 CAD per policy design). Source: Simulation calibrated to reported trends using Alberta public reporting and supporting literature. [3,4,16].

capacity expansion through workforce measures, tele-mental health scale-up, and recovery-oriented infrastructure; values are modeled and not direct measures of system-wide supply.

**5.2.3. Modeled equilibrium comparison.** The modeled intersection of post-intervention supply and demand curves (S₂ ∩ D₂) illustrates a higher but capacity-constrained equilibrium quantity under a zero-copayment environment. Using the baseline calibration:

- Pre-intervention modeled equilibrium: $Q_0 = 60$ services/month at a unit cost proxy of 150 CAD/service

- Demand-implied post-intervention volume: 87 services/month at the same unit cost proxy (150 CAD/service)

- Capacity-constrained post-intervention equilibrium: $Q^* = 78$ services/month at the same unit cost proxy (150 CAD/service), implying ~9 services/month unmet demand (interpreted as potential congestion/queueing rather than additional utilization)

Because Alberta's publicly funded services are designed to maintain zero user copayment, the unit cost proxy is used only to express modeled equilibrium relationships in interpretable units. It should not be interpreted as an observed market price. The modeled stability of the unit cost proxy alongside an increase in equilibrium quantity is consistent with the theoretical expectation that proportional outward shifts in supply and demand can increase utilization without increasing user-facing prices in publicly financed settings [4,6]. However, when demand growth outpaces capacity expansion (as in the base case), the simulation implies a supply-constrained system, in which additional help-seeking does not fully translate into delivered services.

**Implications:** The simulation suggests that coordinated capacity expansion may partially offset demand growth in a zero-copayment setting; however, the magnitude of modeled effects depends on parameter assumptions and should be interpreted directionally, with potential for unmet need if capacity lags. As shown in Figs 4 and 5, the modeled equilibrium quantity increases from $Q_0 = 60$ to $Q^* = 78$ services/month under the illustrative calibration, while the unit-cost proxy is held

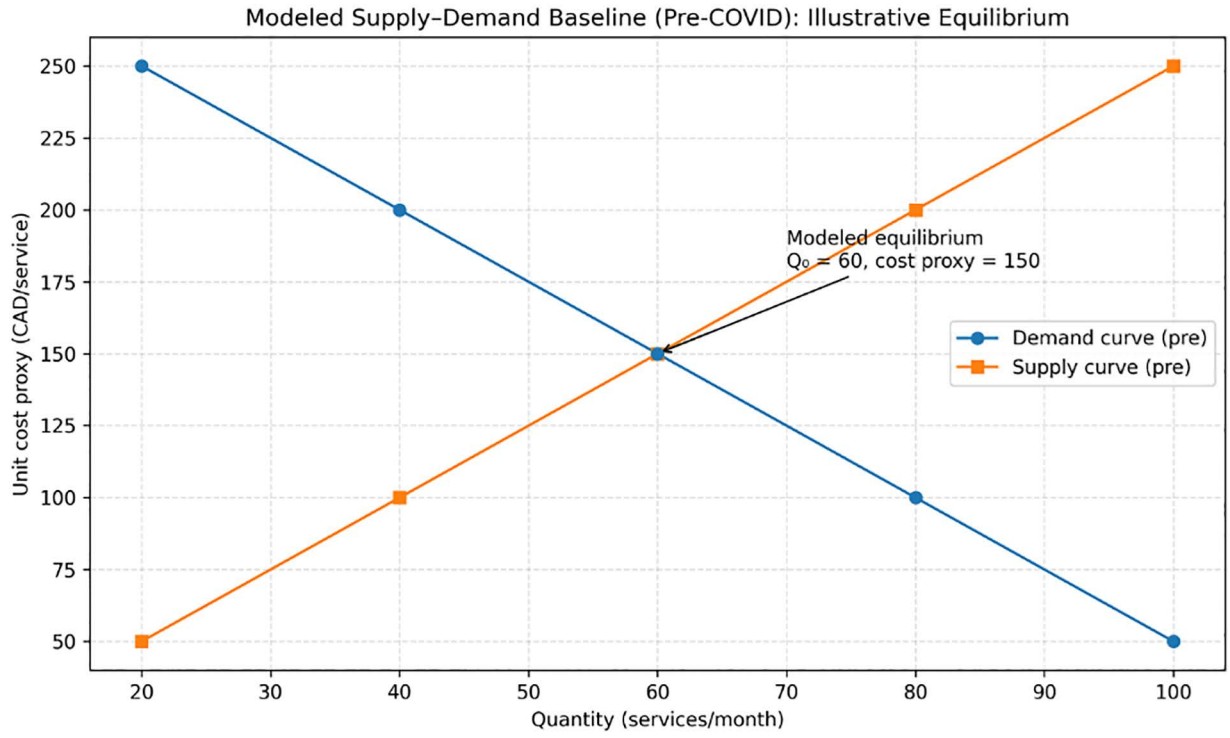

**Fig 4. Pre-COVID mental health service system in Alberta: modeled baseline equilibrium (illustrative).** Note: This figure shows an illustrative, modeled supply–demand equilibrium representing baseline conditions prior to the COVID-19 period ($Q_0 = 60$ services/month). The vertical-axis value (150 CAD/service) is a unit cost proxy used solely to express equilibrium mechanics in interpretable units; it is not an observed market price and does not represent patient copayment (user copayment is 0 CAD by policy design). Source: Simulation calibrated to reported trends using Alberta Annual Report data and supporting literature [2–4].

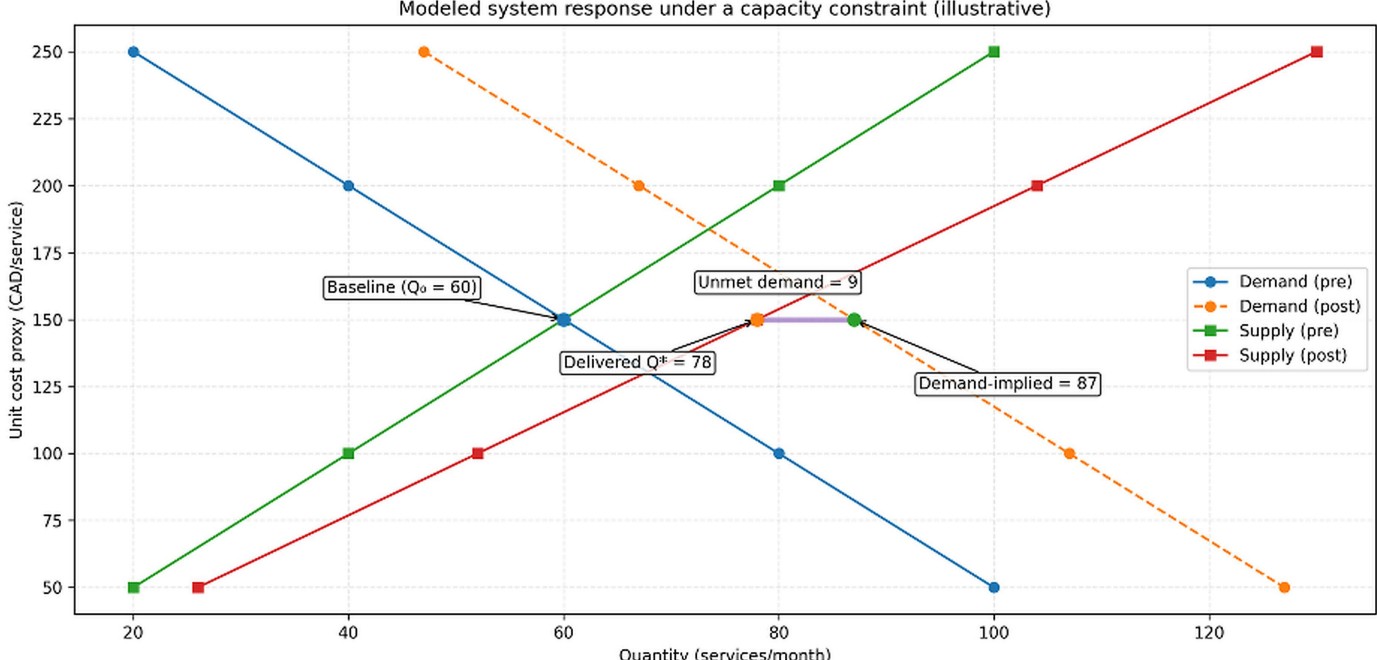

**Fig 5. Post-COVID system response in Alberta: modeled equilibrium shift under public intervention (illustrative).** Note: This figure shows modeled outward shifts in demand and supply intended to illustrate the mechanics of equilibrium after pandemic-era interventions. The modeled post-intervention equilibrium increases to $Q_1 = 78$ services/month, demand to 87, while holding the unit cost proxy constant (150 CAD/service) for illustration; this does not reflect an observed stable market price and does not reflect patient copayment (user copayment remains 0 CAD by policy). Source: Simulation calibrated to reported trends using Alberta Annual Report data and supporting literature [2–4].

constant for visualization; this proxy is not an observed market price and does not represent patient copayments (0 CAD by policy design).

### 5.3. Uncertainty and robustness: deterministic sensitivity and scenario checks

Uncertainty was assessed using deterministic one-way sensitivity analyses on a small set of influential parameters specified a priori: the magnitude of capacity expansion, the magnitude of demand increase, and baseline equilibrium utilization. For each parameter, we applied conservative low–high bounds informed by public reporting when available and by benchmark/guideline sources otherwise, holding other inputs constant [14,19]. As shown in Table 4, the modeled equilibrium quantity ($Q^*$) varies across plausible bounds, indicating that the *magnitude* of the modeled effect is sensitive to assumptions where Alberta-specific parameters are unavailable. Importantly, the sensitivity results also clarify a **capacity-constrained implication**: when demand shifts are large relative to capacity expansion, additional need does not translate into proportionate increases in delivered volume. Instead, the model implies rising **unmet demand**, operationalized as congestion/queueing pressures beyond $Q^*$, reinforcing that expansion may stabilize access only if capacity growth keeps pace with demand growth.

$$Q^* = \min\left(Q_0 + \Delta D,\ Q_0(1 + \Delta C)\right),$$

Where $Q_0$ is the baseline equilibrium quantity, $\Delta D$ is the modeled demand increase (additive), and $\Delta C$ is the modeled capacity expansion (proportional). When $Q_0 + \Delta D > Q_0(1 + \Delta C)$, the system is supply-limited: $Q^*$ reflects deliverable

**Table 4. Deterministic sensitivity analysis of delivered volume (Q*) under a capacity-constrained system (illustrative).**

**A. One-way sensitivity (one parameter varied; others held at base).**

*One parameter varied at a time; all others held at base values.*

| Parameter varied | Base | Low | High | Delivered Q* at base | Delivered Q* at low | Delivered Q* at high | Interpretation |
|---|---|---|---|---|---|---|---|
| Capacity expansion (system) | 30% | 20% | 40% | 78 | 72 | 84 | When **capacity is the binding constraint**, increasing capacity raises the *deliverable* volume (Q*) up to the level of demand; beyond that, additional capacity would not increase Q* unless demand also rises. |
| Demand increase (system) | +27 | +20 | +35 | 78 | 78 | 78 | When **capacity binds**, higher demand does **not** raise Q*; it increases the **unmet demand/queue** (gap between demand-implied volume and deliverable volume). |
| Baseline equilibrium quantity ($Q_0$) | 60 | 55 | 65 | 78 | 72 | 85 | Changes in $Q_0$ shift both demand-implied and capacity-implied volumes; Q* remains determined by the **minimum** of the two (capacity-limited when demand exceeds deliverable supply). |

**B. Multi-way scenario checks (joint variation of capacity and demand).**

| Scenario | Capacity expansion | Demand increase | Delivered Q* | Interpretation |
|---|---|---|---|---|
| Capacity-limited (low capacity + high demand) | 20% | +35 | 72 | Congestion/unmet demand likely (demand exceeds deliverable capacity). |
| Demand-limited (high capacity + low demand) | 40% | +20 | 80 | Capacity sufficient; delivered volume limited by demand. |

*Note:* Q* denotes the modeled **delivered** service volume (services/month) under the illustrative supply–demand framework in a zero-copayment setting. Delivered volume is computed as:*

*volume and the remainder is interpreted as unmet demand (queueing/congestion) rather than additional utilization. Ranges are conservative and intended as directional robustness checks consistent with modeling and reporting guidance [14,19]. Values are rounded to whole services.*

## 6. Discussion

### 6.1. Interpreting Alberta's system response in context

Alberta's pandemic-era expansion of publicly funded mental health services illustrates how policy action can support access under a zero-copayment design. Empirically, program delivery met or exceeded planned targets for several initiatives, suggesting strong implementation performance during 2023–2024 [3]. In the illustrative supply–demand simulation, outward shifts in modeled demand and capacity increased the modeled equilibrium quantity (Figs 2–5), consistent with the economic expectation that capacity expansion can partially offset increases in service use when financial barriers are low [4,6]. These outputs are simulations calibrated to reported trends and should be interpreted as mechanisms rather than observed prices or system-wide utilization counts.

### 6.2. Economic frictions and ethical trade-offs

Alberta's approach is plausibly responsive to well-described frictions in health care markets, including moral hazard under low user cost and information asymmetry [4,6]. It also occurs in a broader context of persistent unmet need and access gaps documented across jurisdictions [12,13]. Supplier-induced demand is a relevant behavioral consideration

when clinical discretion and service intensity interact with incentives and monitoring capacity; in this study, it is most appropriate as a rationale for sensitivity analysis around provider behavior rather than as evidence of overuse in Alberta [7].

Equity–efficiency trade-offs remain. Expansion can still yield uneven benefits if lower-acuity users in better-served areas absorb new capacity while rural or remote communities face structural barriers to access [13]. Even with zero user copayments, opportunity costs persist—including workforce strain, congestion, and possible diversion from other services—risks that are difficult to quantify solely through public reporting [3,16].

### 6.3. Comparative insight and transferability

Relative to more centralized coordination and performance-monitoring approaches (e.g., Ontario Health's Mental Health and Addictions Centre of Excellence), Alberta's programmatic expansion may enable faster rollout but may leave gaps in standardized outcome tracking and equity measurement [8]. Internationally, tele-mental health scale-up has been widespread, yet uptake inequities related to digital access and literacy remain common across settings [5,9,10]. Alberta's experience is consistent with post-pandemic calls for integrated systems that combine digital access with community-based pathways [5]. Sustained gains likely depend on strengthening data infrastructure, performance measurement, and equity-focused implementation over time [5,8,10].

Taken together, Alberta's experience suggests that rapid capacity expansion can partially accommodate pandemic-era demand pressures in a publicly financed, zero-copayment context, but that sustainability depends on workforce, equity in access, and performance infrastructure. Broader national and global guidance similarly emphasizes that mental health system strengthening requires durable financing, integrated community and digital pathways, and accountability mechanisms that make access and outcomes visible over time [17,20–22].

## 7. Policy implications and recommendations

1. **Workforce capacity and distribution.** Sustaining post-crisis access will likely require workforce strategies that address geographic maldistribution (e.g., rural/remote), stabilize recruitment and retention, and support supervised practice pathways for multidisciplinary teams—paired with explicit service targets and monitoring [3,16].

2. **Telehealth as a permanent modality, with equity safeguards.** Tele-mental health can extend reach, but equitable benefit depends on broadband availability, provider competency in virtual care, and patient-facing supports (digital literacy, navigation, language accessibility). Monitoring should explicitly track who is and is not reached by virtual pathways [5,9,10].

3. **Sustainable financing and governance beyond "surge" dynamics.** If crisis-period funding enabled rapid rollout, longer-term performance depends on clearer recurrent financing arrangements and governance mechanisms that prevent volatility, define accountability, and protect equity objectives (e.g., explicit reporting requirements and outcome expectations) [20–22].

4. **Equity-focused implementation and culturally safe pathways.** Because access gaps persist even in publicly funded systems, reforms should include explicit equity design features (e.g., navigation supports, culturally safe pathways, and community partnerships where appropriate) and the measurement of distributional effects (e.g., by geography and other policy-relevant equity markers) [12,13,20].

5. **Data-driven performance monitoring and transparency.** Decentralized expansion benefits from centralized measurement capacity: standardized indicators, dashboards, and routine public reporting that link service activity to access, timeliness, and outcomes. Ontario's emphasis on coordination and monitoring illustrates one model of system-level performance infrastructure [8].

6. **Demand activation paired with throughput capacity.** Public awareness and stigma-reduction efforts are most defensible when paired with clear pathways to care (triage/referral clarity, wait-time transparency, stepped-care options), so increased demand does not simply translate into congestion [5,20–22].

## 8. Limitations

This analysis relies on publicly available reports and secondary sources rather than patient-level administrative data. As a result, program reporting does not consistently provide standardized, disaggregated outcomes (e.g., symptom change, functional improvement, patient-reported experience), equity-stratified utilization, or comparable effectiveness metrics across initiatives [3]. Cost and economic inputs were also constrained: Alberta-specific marginal costs by service type, demand elasticities, and longer-term cost offsets were not available from public sources and were therefore represented using conservative ranges and bounded assumptions informed by general health economics and related evidence [4,6,17]. These choices support plausibility checks but limit causal inference and precision.

Second, the empirical component focuses on a single fiscal year (2023–2024), which restricts inference about longer-term sustainability, workforce dynamics, and whether post-crisis utilization patterns persist as service models mature [3,16]. Third, the simulation is intentionally illustrative: it translates reported trends into an interpretable mechanism-based narrative but should not be read as an estimate of observed prices, system-wide utilization totals, or welfare impacts. Consistent with modeling good-practice guidance, the simulation is best interpreted as a structured sensitivity framework rather than a predictive or causal model [14,15]. Also, Public reporting lacks stratified outcomes by equity groups (e.g., Indigenous: 43% recovery capital improvement in MRP, p.20), limiting disaggregated analysis. Finally, we did not estimate program costs, QALYs, ICERs, or budget impact; these analyses were not feasible given the limits of public reporting [18,19].

## 9. Future research

Future work should prioritize five directions:

1. **Patient-level, longitudinal evaluation.** Link administrative utilization with outcomes and costs over time to support ROI, budget impact, and longer-horizon assessments where feasible, consistent with Canadian economic evaluation guidance and reporting standards [18,19].

2. **Equity-focused measurement.** Conduct equity audits that map access, timeliness, and outcomes across rural/remote communities and priority populations, using geographic and demographic stratification, and reflect known cross-jurisdictional disparities [12,13].

3. **Program-specific effectiveness beyond volumes.** Evaluate CASA, VODP, recovery communities, and tele-mental health using outcomes beyond service counts (e.g., clinical outcomes, continuity, relapse/retention, patient experience), building on the limitations of volume-based reporting [3,5].

4. **Workforce sustainability.** Examine recruitment/retention, workload, burnout, and skill-mix evolution to test whether capacity gains can be maintained without quality erosion [16].

5. **Digital equity and implementation design.** Quantify barriers to tele-mental health uptake (connectivity, digital literacy, privacy constraints) and test mitigation strategies, drawing on emerging evidence that virtual care can reproduce inequities without targeted supports [5,9,10].

Comparative evaluations across provinces could also clarify how governance and performance infrastructure shape equity and accountability during rapid scale-up—particularly contrasts between more centralized monitoring approaches and decentralized program expansion [8].

## 10. Conclusion

Alberta's 2023–2024 mental health system response suggests that coordinated, publicly funded interventions can expand *reported* service delivery during system shocks. Public reporting indicates that several major initiatives met or exceeded planned delivery targets during the period examined [3]. The accompanying illustrative supply–demand simulation offers a mechanism-based interpretation in which concurrent increases in capacity and help-seeking raise the modeled equilibrium quantity under a zero-copayment design; modeled values should not be interpreted as observed prices or system-wide utilization counts. Sustaining gains likely depends on workforce stabilization, digital equity supports, culturally safe and navigable care pathways, and strengthened performance monitoring so that expansion translates into equitable access rather than uneven uptake or congestion.

## Supporting information

**S1 Text. README for OSF Project.**
(PDF)

**S1 File. Data Assumption.**
(XLSX)

**S1 Checklist. CHEERS Checklist.**
(DOCX)

**S1 Fig. Graphical abstract: Alberta's mental health system response (2023–2024).** This graphical abstract summarizes key publicly funded interventions and empirically reported program delivery outputs from Alberta's 2023–2024 reporting, alongside an illustrative supply–demand framework depicting the directional mechanics of equilibrium under a zero-copayment design. Modeled quantities shown in the equilibrium pathway are simulations calibrated to reported trends and should not be interpreted as observed prices or system-wide utilization.
(TIF)

## Author contributions

**Conceptualization:** Kola Adegoke, Abimbola Adegoke.

**Data curation:** Deborah Dawodu, Temitope Kayode, Mallika Singh.

**Formal analysis:** Kola Adegoke, Akorede Adekoya.

**Funding acquisition:** Olajide Alfred Durojaye, Abiodun Isola Aluko.

**Investigation:** Deborah Dawodu, Ayoola Bayowa, Akorede Adekoya.

**Methodology:** Deborah Dawodu, Ayoola Bayowa, Temitope Kayode.

**Project administration:** Kola Adegoke.

**Resources:** Akorede Adekoya, Mallika Singh.

**Software:** Kola Adegoke.

**Supervision:** Kola Adegoke, Abimbola Adegoke.

**Visualization:** Ayoola Bayowa.

**Writing – original draft:** Kola Adegoke.

**Writing – review & editing:** Abimbola Adegoke, Temitope Kayode, Mallika Singh, Olajide Alfred Durojaye, Abiodun Isola Aluko, Adeyinka Adegoke.

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
