## [Decision Letter · Decision Letter 0]

15 Jul 2025

PMEN-D-25-00038

From Crisis to Recovery: Exploring the Demand Surge for Mental Health Services in Alberta, Canada

PLOS Mental Health

Dear Dr. Kola Adegoke,

Thank you for submitting your manuscript to PLOS Mental Health. After careful consideration, we feel that it has merit but does not fully meet PLOS Mental Health’s publication criteria as it currently stands. Therefore, we invite you to submit a revised version of the manuscript that addresses the points raised during the review process.

Please submit your revised manuscript by July 29th. If you will need more time than this to complete your revisions, please reply to this message or contact the journal office at mentalhealth@plos.org. Please include the following items when submitting your revised manuscript:

We look forward to receiving your revised manuscript.

Kind regards,

Olumide Thomas Adeleke, MBBS, FWACP

Academic Editor

PLOS Mental Health

Journal Requirements:

2. We have amended your Competing Interest statement to comply with journal style. We kindly ask that you double check the statement and let us know if anything is incorrect.

4. We ask that a manuscript source file is provided at Revision. Please upload your manuscript file as a .doc, .docx, .rtf or .tex.

5. Please provide separate figure files in .tif or .eps format.

https://journals.plos.org/mentalhealth/s/figures

https://journals.plos.org/mentalhealth/s/figures#loc-file-requirements

6. Please upload a copy of Figure 1 which you refer to in your text on page 9. Or, if the figure is no longer to be included as part of the submission please remove all reference to it within the text.

Reviewers' comments:

Reviewer's Responses to Questions

**Comments to the Author**

1. Does this manuscript meet PLOS Mental Health’s publication criteria?

Reviewer #1: Yes

Reviewer #2: No

2. Has the statistical analysis been performed appropriately and rigorously?

Reviewer #1: I don't know

Reviewer #2: Yes

3. Have the authors made all data underlying the findings in their manuscript fully available (please refer to the Data Availability Statement at the start of the manuscript PDF file)?

Reviewer #1: Yes

Reviewer #2: No

4. Is the manuscript presented in an intelligible fashion and written in standard English?

Reviewer #1: Yes

Reviewer #2: Yes

Reviewer #1: Introduction

Missing citation. Please site appropriately

“Alberta's 2023-24 Annual Report reveals significant investments in expanding mental health services, such as adding over 10,000 publicly funded addiction treatment spaces and eliminating user fees for live-in addiction treatment programs”.

“These efforts were supported by innovative telehealth solutions like Virtual Opioid Dependency Program (VODP), which offered same-day opioid dependency treatment to over 16,000 Albertans”

Under Systemic Gaps please change “linking to linkage”

Apart from that, inadequate funding for community-based interventions and a lack of linking between primary care and mental health services have been identified as systemic gaps in mental

Under the subheading Stable Equilibrium Before Change

Missing citation. Please site appropriately “

“According to health economics principles, a stable equilibrium reflects a balance between supply and demand, constrained by existing infrastructure and policy limitations”.

Under the subheading Barriers to access

Missing citation to support this claim. Please site appropriately.

“Youths, particularly those from low-income families, were disproportionately affected by the lengthy wait times for publicly funded services, which often exceeded one month”.

Results and Discussion

Consider having a separate results and discussion rather than combining. Distinct results and discussion headings will help the readership and also help demarcate where the results start and the discussion begin.

Discussion may include comparison of what is happening in other provinces. At least one province is recommended.

Reviewer #2: This manuscript examines how Alberta, Canada responded to increased demand for mental health services during the COVID-19 pandemic through various policy interventions. The authors apply economic principles, specifically supply and demand curve analysis, to evaluate how government initiatives affected market equilibrium, service accessibility, and pricing. The paper describes programs like CASA Mental Health, Virtual Opioid Dependency Program (VODP), and recovery communities, analyzing their impact on both demand and supply curves for mental health services.

Strengths

1. Timely and relevant topic: The paper addresses the critical issue of mental health service provision during and after a global pandemic, which has significant public health implications.

2. Clear economic framework: The authors effectively apply economic principles to analyze healthcare policy interventions, providing a useful lens for understanding market dynamics.

3. Data-driven approach: The paper incorporates real-world data from Alberta's 2023-24 Mental Health and Addiction Annual Report, strengthening the analysis with concrete figures on service expansion and usage.

4. Visual representation: Figure 1 effectively illustrates the shifting demand and supply curves, making the economic concepts accessible to readers.

5. Practical policy implications: The authors provide specific, actionable recommendations for workforce development, telehealth expansion, sustainable funding, and culturally appropriate services.

Methodological Concerns

1. Data sources and quality: While the paper mentions using data from government reports, there needs to be more transparency about how the specific values in Tables 1 and 2 were derived. Were these empirically measured from market data, or are they hypothetical values to illustrate the concept? This distinction should be clearly stated.

2. Assumptions: The equilibrium analysis makes several implicit assumptions that should be explicitly stated and justified, such as:

- The uniformity of services across providers

- The proportional increase in supply across different regions

- The stability of service quality despite rapid expansion

3. Counterfactual analysis: The paper would benefit from considering what might have happened without these interventions, particularly given the pandemic-driven increase in mental health needs.

Theoretical Framework

1. Beyond simple supply-demand: While the basic economic framework is well-applied, the analysis could be enriched by incorporating concepts specific to healthcare economics such as:

- Moral hazard

- Adverse selection

- Information asymmetry

- Provider-induced demand (mentioned but not fully explored)

2. Equity considerations: Although the paper mentions rural and Indigenous populations, a more robust analysis of how these interventions address or fail to address pre-existing inequities would strengthen the paper.

Empirical Evidence

1. Outcome measurements: The paper focuses primarily on service provision (supply) and utilization (demand), but lacks analysis of clinical outcomes or quality indicators. Did increased access translate to improved mental health outcomes?

2. Cost-effectiveness: The revenue example is helpful but insufficient. A more comprehensive analysis of cost-effectiveness would strengthen the policy implications.

3. Long-term sustainability: While the paper mentions the need for sustainable funding models, a more detailed analysis of potential funding mechanisms would be valuable.

Structural Issues

1. Literature context: The paper would benefit from a more comprehensive literature review situating Alberta's approach within broader research on mental health policy responses to the pandemic.

2. Abstract clarity: The abstract repeats itself in the first and second paragraphs, which should be corrected.

3. Tables and figures: While Figure 1 is helpful, additional visualization of real outcome data (not just the theoretical curves) would strengthen the paper's impact.

**Do you want your identity to be public for this peer review?** For information about this choice, including consent withdrawal, please see our Privacy Policy

Reviewer #1: No

Reviewer #2: No

---

## [Decision Letter · Decision Letter 1]

28 Dec 2025

PMEN-D-25-00038R1

From Crisis to Recovery: Exploring the Demand Surge for Mental Health Services in Alberta, Canada

PLOS Mental Health

Dear Dr. Adegoke,

Thank you for submitting your manuscript to PLOS Mental Health. After careful consideration, we feel that it has merit but does not fully meet PLOS Mental Health’s publication criteria as it currently stands. Therefore, we invite you to submit a revised version of the manuscript that addresses the points raised during the review process.

Consider the constructive comments from the reviewers (comments from reviewer 5 is attached in a file). In addition, consider: 1) subsection '4.5 Limitations' is not part of the methods, but rather part of the discussion; 2) check the overuse of bullet points; and 3) mainly, the Results section appears to combine observed empirical findings with synthetic, model-based results derived from supply–demand simulations. I recommend reorganizing this section into clearly labeled subsections (e.g., empirical vs. modeled results) to improve transparency and avoid potential misinterpretation of simulated values as directly observed data.

We look forward to receiving your revised manuscript.

Kind regards,

Ariel Soares Teles

Academic Editor

PLOS Mental Health

Journal Requirements:

Additional Editor Comments (if provided):

Reviewers' comments:

Reviewer's Responses to Questions

**Comments to the Author**

Reviewer #3: (No Response)

Reviewer #4: (No Response)

Reviewer #5: (No Response)

publication criteria?

Reviewer #3: Partly

Reviewer #4: Yes

Reviewer #5: Yes

3. Has the statistical analysis been performed appropriately and rigorously?

Reviewer #3: Yes

Reviewer #4: Yes

Reviewer #5: Yes

4. Have the authors made all data underlying the findings in their manuscript fully available (please refer to the Data Availability Statement at the start of the manuscript PDF file)?

Reviewer #3: No

Reviewer #4: Yes

Reviewer #5: (No Response)

5. Is the manuscript presented in an intelligible fashion and written in standard English?

Reviewer #3: Yes

Reviewer #4: Yes

Reviewer #5: Yes

Reviewer #3: The study’s intent and direction are appropriate for PLOS Mental Health. To reach publishable standards, several transparency and reproducibility items must be addressed.

Major points

Data & Code Availability (Required). Please deposit all data underlying the findings (including the processed data behind means/medians and any figure values) and the analysis files/code in a public repository (e.g., Zenodo/OSF/Dryad) and update the Data Availability Statement with a DOI. Include a README with step-by-step instructions to reproduce the tables and figure.

Methods Reporting (Strengthen). Expand the Methods to specify: analytic perspective (payer/societal), time horizon, base currency and price year, any inflation/PPP adjustments applied, and software + version used. Provide a table with all parameter values and their sources. If your horizon exceeds one year, state and justify the discount rate; if one-year only, state discounting is not applicable.

Uncertainty & Robustness. Add at least a structured deterministic sensitivity analysis (e.g., one-way ranges on key parameters; a tornado diagram if space allows). Consider a scenario analysis (e.g., alternative uptake/implementation paths). Clearly state how uncertainty affects the interpretation of budget implications.

Results vs. Discussion. For readability and alignment with journal expectations, please separate Results and Discussion. Keep quantitative outputs in Results; place interpretation, policy implications, and limitations in Discussion.

Figures & Tables. Ensure the figure cited in the text is provided as a separate figure file upon submission (with an appropriate caption in the manuscript body). For tables, include units, denominator definitions, and the price year in the titles or footnotes.

Reporting Guidelines. Given the economic focus, please complete and submit the CHEERS 2022 checklist as Supporting Information, mapping each checklist item to a manuscript location.

Ethics Statement. Although no human participants were involved, add a brief statement clarifying that only publicly available, non-identifiable data were used and that institutional ethics approval/consent were not required.

Use of AI/LLM (Transparency). If any AI tools supported writing/formatting/translation, please add a short statement describing tool, purpose, and human verification. If none were used, state so.

References (Journal Style). Convert in-text citations to numerical Vancouver style and format the reference list accordingly (include DOIs where available).

Minor points

Standardize terminology (e.g., “budget impact” vs “economic assessment”) and define all abbreviations at first mention.

Check minor copyedits (spacing after “Keywords:”; consistent capitalization; tense consistency).

Consider adding a brief Limitations subsection (e.g., reliance on publicly reported budget lines, uncertainty in implementation timelines, lack of indirect cost capture if perspective is payer).

Where relevant, report currency with symbol and ISO code (e.g., CAD) and include the price year alongside amounts.

Reviewer #4: Manuscript Review Report

GENERAL COMMENTS

would like to sincerely commend you for this well-crafted and timely manuscript. The study explores the surge in demand for mental health services in Alberta, Canada, during the COVID-19 pandemic, focusing on government interventions, public awareness, and the role of recovery programs. The integration of demand-supply economic analysis offers a unique and robust framework for understanding the dynamics of service provision and accessibility in a public health crisis. Below are further suggestions intended to strengthen the manuscript.

1. Your abstract is comprehensive but could benefit from a more actionable conclusion that directly emphasises policy implications. Example: “The findings suggest that the government’s targeted interventions, including CASA Mental Health and VODP, significantly increased service access while stabilising service prices. However, long-term sustainability requires continued investment in workforce development, telehealth infrastructure, and public-private partnerships.”

2. Your introduction could be more explicit in linking the study’s focus to broader global mental health policy trends.

3. Clarifying the specific policy frameworks and economic theories used to underpin the demand-supply analysis could further strengthen the introduction’s connection to the broader research and policy landscape.

4. Explaining how external factors (such as population growth or economic shifts) were accounted for, or why they were excluded, would add further rigour to the analysis.

5. Including a brief note on how geographic variability (e.g., urban vs rural, different countries) might influence generalizability would be useful.

6. The discussion thoughtfully contextualises the findings, but it could be enhanced by a deeper exploration of how the demand-supply dynamics directly inform policy decisions beyond Alberta.

7. You might expand the discussion of public-private partnerships, particularly in sustaining long-term funding and innovation for mental health services.

8. The manuscript is methodologically rigorous, but limitations of the review process could be more explicitly acknowledged, particularly regarding potential publication bias.

9. Consider providing more specific recommendations for future research based on identified gaps.

Reviewer #5: All comments will be found in the word document attached. I thank the authors for their submission.

**Do you want your identity to be public for this peer review?** For information about this choice, including consent withdrawal, please see our Privacy Policy

Reviewer #3: No

Reviewer #4: No

Reviewer #5: No

---

## [Decision Letter · Decision Letter 2]

19 Feb 2026

From Crisis to Recovery: Exploring the Demand Surge for Mental Health Services in Alberta, Canada — A document-based policy analysis with an illustrative supply–demand simulation (2023–2024)

PMEN-D-25-00038R2

Dear Dr Adegoke,

We are pleased to inform you that your manuscript 'From Crisis to Recovery: Exploring the Demand Surge for Mental Health Services in Alberta, Canada — A document-based policy analysis with an illustrative supply–demand simulation (2023–2024)' has been provisionally accepted for publication in PLOS Mental Health.

Best regards,

Ariel Soares Teles

Academic Editor

PLOS Mental Health

Reviewer Comments (if any, and for reference):

Reviewer's Responses to Questions

**Comments to the Author**

Reviewer #3: All comments have been addressed

Reviewer #5: All comments have been addressed

Reviewer #6: All comments have been addressed

publication criteria?

Reviewer #3: Yes

Reviewer #5: Yes

Reviewer #6: Yes

3. Has the statistical analysis been performed appropriately and rigorously?

Reviewer #3: Yes

Reviewer #5: Yes

Reviewer #6: Yes

4. Have the authors made all data underlying the findings in their manuscript fully available (please refer to the Data Availability Statement at the start of the manuscript PDF file)?

Reviewer #3: Yes

Reviewer #5: Yes

Reviewer #6: Yes

5. Is the manuscript presented in an intelligible fashion and written in standard English?

Reviewer #3: Yes

Reviewer #5: Yes

Reviewer #6: Yes

Reviewer #3: Dear authors, I suggest paying special attention to the following two points in order to avoid issues during the technical check:

Competing interests: inconsistency between the submission form and the manuscript

The submission form states: “The authors have declared that no competing interests exist.” However, the manuscript reports an affiliation/relationship with CASA Mental Health as a potential competing interest or relevant relationship. I recommend aligning both declarations so that the Competing Interest statement in the system and in the manuscript are identical, or at least fully consistent.

Figure technical requirements (format).

Although the figure appears embedded in the document, PLOS requires figures to be uploaded as separate files in .tif or .eps format. The fact that figures appear in the PDF does not confirm that they were uploaded in the required format. I recommend verifying that all figures (including Figure 1) have been uploaded in the requested formats to avoid the revision being flagged as “incomplete” during the technical check.

Reviewer #5: Most of the issues I previously mentioned have been resolved. The overall focus of the paper is more clear, and this is reflected in better written results and discussion sections. The introduction also structures the paper much better than previously. The literature review section has incorporated a lot of important variables, but could do with a bit more organisation. For example, signposting as to what you are going to cover and why. I think the same of the results section, although these are significantly more clear than previous. However, I am not an economist (I am a psychologist and public health scientist) or well positioned to address the data analysis and presentation of results portions of the paper. I believe othere reviewers will be better suited to comment on these sections. Overall however; the paper looks much better and any further edits on behalf of the authors do not require additional reviews, in my opinion.

Reviewer #6: (No Response)

**Do you want your identity to be public for this peer review?** For information about this choice, including consent withdrawal, please see our Privacy Policy

Reviewer #3: No

Reviewer #5: No

Reviewer #6: No
